# JmjC domain-containing histone demethylase gene family in Chinese cabbage: Genome-wide identification and expressional profiling

**Fengrui Yin**[1☯], **Yuanfeng Hu**[2☯], **Xiaoqun Cao**[1], **Xufeng Xiao**[1]*, **Ming Zhang**[3], **Yan Xiang**[1], **Liangdeng Wang**[1], **Yuekeng Yao**[1], **Meilan Sui**[1], **Wenling Shi**[3]

**1** College of Agronomy, Jiangxi Agricultural University, Nanchang, Jiangxi Province, P. R. China,
**2** Agricultural Sciences Research Center, Pingxiang, Jiangxi Province, P. R. China, **3** Department of Biotechnology, Jiangxi Biotech Vocational College, Nanchang, Jiangxi Province, P. R. China

☯ These authors contributed equally to this work.
* xiaoxf@jxau.edu.cn

**Data Availability Statement:** All relevant data are within the manuscript and its Supporting Information files.

## Abstract

The Jumonji C (JmjC) structural domain-containing gene family plays essential roles in stress responses. However, descriptions of this family in *Brassica rapa* ssp. *pekinensis* (Chinese cabbage) are still scarce. In this study, we identified 29 members of the *BrJMJ* gene family, with cis-acting elements related to light, low temperature, anaerobic conditions, and phytohormone responses. Most *BrJMJ*s were highly expressed in the siliques and flowers, suggesting that histone demethylation may play a crucial role in reproductive organ development. The expression of *BrJMJ1*, *BrJMJ2*, *BrJMJ5*, *BrJMJ13*, *BrJMJ21* and *BrJMJ24* gradually increased with higher Cd concentration under Cd stress, while *BrJMJ4* and *BrJMJ29* could be induced by osmotic, salt, cold, and heat stress. These results demonstrate that *BrJMJ*s are responsive to abiotic stress and support future analysis of their biological functions.

## Introduction

In eukaryotes, each nucleosome comprises 146 base pairs of DNA wrapped around a histone octamer, which includes a single H3–H4 tetramer and two H2A–H2B dimers. Besides their spherical structural domain, histones feature a flexible, charged amino-terminal (N-terminal) tail [1,2]. Histone post-translational modifications (PTMs) regulate chromatin structure through processes such as methylation, ubiquitination, and acetylation, particularly at N-terminal tail residues like lysine, arginine, and serine [3–5]. The addition or removal of these chemical groups on specific histone residues may alter histone–DNA interactions. These modifications are recognized as signals by chromatin-modifying proteins, thereby regulating transcriptional activities [4].

Methylation, one of the most common PTMs, primarily occurs at lysine and arginine residues and is catalyzed by protein arginine methyltransferases (PRMTs) and Suppressor of variegation, Enhancer of zeste, Trithorax (SET) structural domain proteins. Histone methylation is

**Funding:** The Natural Science Foundation of China (31860560) and the Natural Science Foundation of Jiangxi Province (20224BAB205027). The funders had no role in study design, data collection and analysis, decision to publish, or preparation of the manuscript.

usually reversible and can be regulated by methyltransferases and demethylases. There are two key demethylases. The first is lysine-specific demethylase 1 (LSD1, also known as KIAA0601), a nuclear ortholog of amine oxidase and the first demethylase to be discovered. The second is JmjC structural domain-containing histone demethylase (JHDM), which maintains histone methylation homeostasis in vivo [2,6,7]. As a histone demethylase, LSD1 demethylates lysine via a formaldehyde oxidation reaction [8,9]. Jumonji domain-containing proteins (JmjC) were initially identified in mice via a gene-trapping strategy that produced mutations in a gene critical for normal neural tube morphogenesis. The name "Jumonji" (meaning "cross" in Japanese) was derived from changes in neural plate morphology in these mutant mice [10]. Structural analysis has revealed that Jumonji, in the dioxygenase superfamily, has a conserved structural domain (the JmjC domain) with conserved 2-oxoglutarate-Fe (II) binding sites [11]. Proteins containing the JmjC structural domain may reverse histone methylation, while the substitution of histidine residues essential for Fe (II) interactions disrupts their demethylase activity. This suggests that demethylase activity depends on the integrity of the JmjC structural domain [12].

In animals, many key methylated histone marks with corresponding demethylases have been identified. Several histone demethylases with only the JmjC structural domain (referred to as "JmjC domain-only groups") have been classified based on sequence similarity: lysine-specific demethylase 2 (KDM2)/JHDM1/FBX, KDM3/JMJD1/JHDM2, KDM4/JMJD2, KDM5/JARID, and JMJD3/KDM6. These groups target particular histone lysines with varying methylation statuses [13–15]. Although most JmjC proteins in animals are conserved in plants, *Arabidopsis thaliana* (*Arabidopsis*) and rice (*Oryza sativa* L.) lack the KDM2/JHDM1 and KDM6/MJD motifs responsible for demethylating H3K36me2/1 and H3K27me3/2, respectively. In plants, *JmjC* family members are divided into nine subgroups: JMJ6, KDM3/JHDM2, KDM5/JARID1, putative KDMs (PKDMs) PKDM7–PKDM9, and PKDM11–PKDM13 [6,11,16]. In JmjC proteins, mutations in structural domains containing divalent ferrous ions, α-ketoglutarate, and histone polypeptide sites can significantly affect their catalytic activity [6]. For instance, in *Rosa Chinensis* Jacq. (Chinese rose), the lack of histone demethylase activity in *RcJMJ40* is presumed to be due to the absence of two divalent ferrous ions and a fragment of the α-ketoglutarate binding site [14].

In plants, JmjC-containing proteins mediate the epigenetic processes associated with growth and development, transition to flowering, and stress responses [17–19]. *JmjC* gene family members are found in multiple species, including *Arabidopsis* [20], maize (*Zea mays* L.) [21], rice [5], Chinese rose [14]. In *Arabidopsis*, *AtJMJ30* is responsible for regulating flowering time and root growth [22–24]. *AtJMJ16* and *AtJMJ17* are associated with leaf senescence and the osmotic response [25,26]. In *Gossypium hirsutum* (Upland cotton), *GhJMJ34* and *GhJMJ40* significantly enhance the response of the Upland cotton to salt and osmotic stress [27]. In rice, *JMJ706* specifically demethylates the H3K9me2 site and is involved in the regulation of rice flower development [28]. And other *JmjC* genes of rice may play key roles in epigenetic regulation [6]. In summary, the *JmjC* gene may play an important role in plant growth, development and abiotic stress. To further understand their function, it is essential to identify and classify *JmjC* family members and develop sequence-based prediction of their demethylase activity.

Chinese cabbage (*Brassica rapa* ssp. *pekinensis*) [29], the main leafy vegetable in China, with a long cultivation history, a wide variety of species, and distinct morphological characteristics [30]. Abiotic stresses including heat, freezing, cold, salinity, drought and nutrient imbalance, severely affect growth and development of Chinese cabbage [31]. The significance of JmjC structural domain-containing histone demethylases in stress resistance in plants is well recognized. However, the systematic research of the histone demethylase gene family in Chinese cabbage is still lacking, and their specific biological functions remain unclear. This study aimed to comprehensively identify histone demethylase-associated genes in Chinese cabbage

through genome-wide analyses. By analysing gene structure, conserved structural domains, cis-acting element, chromosomal distribution, and gene duplication, we characterized the *JmjC* gene family in Chinese cabbage. Additionally, using quantitative real-time PCR (qRT-PCR), we examined the expression of *JmjC* family member in the leaves under abiotic stress and characterized their evolutionary relationships and abiotic stress response patterns. Therefore, this study may provide a theoretical basis for further research on the functions of *BrJMJ* genes and for improving abiotic stress tolerance in Chinese cabbage.

## Materials and methods

### Plant materials

Seedlings of the Chinese cabbage cultivar 'Xiayang Early 50' (a highly adaptable inbred variety) were planted in the seedling room of the Department of Horticulture, Jiangxi Agricultural University, China. Seedlings with 2.0–3.0 true leaves were transplanted into Hoagland nutrient solution for 10 days. A cadmium (Cd)-containing reagent ($CdCl_2 \cdot 2.5H_2O$) was added to the nutrient solution to achieve varying final Cd concentration of Cd2 (2.0 mg·$L^{-1}$), Cd4 (4.0 mg·$L^{-1}$), Cd6 (6.0 mg·$L^{-1}$), Cd8 (8.0 mg·$L^{-1}$), and Cd10 (10 mg·$L^{-1}$). A Hoagland solution without Cd was used as the control (CK). The treatments lasted for 7.0 days. In addition, other forms of abiotic stress were also assessed, including: osmotic strss (20% PEG6000), salt (200 mM NaCl), cold (4.0°C), and heat (38°C). Among them, osmotic stress (20% PEG6000) and salt (200 mM NaCl) were hydroponic in the seedling room. Hydroponic was performed in the incubator under cold (4.0°C) and heat (38°C) stress. Leaf samples were collected at 0.0, 3.0, 6.0, and 9.0 hours after treatment, immediately frozen in liquid nitrogen, and stored at −80°C for subsequent RT–qPCR assay.

### Genome-wide identification of *BrJMJ* genes

Genomic data for the '*Brara_Chiifu*_V3.0' reference genome, including genome sequences, annotation files, protein sequence files and coding sequences (CDS), were obtained from the Brassicaceae Database (BR-AD) (http://brassicadb.cn/). The *JmjC* genes were downloaded from the *Arabidopsis* Genome Database (TAIR http://ara-bidopsis.org/) and aligned with JmjC protein sequences from the Chinese cabbage protein sequence set using BlastP. Sequences with an e-value of $1 \times 10^{-10}$ were selected using TBtools [30]. Gene function identification was performed using a Hidden Markov Model. *BrJMJ* genes were screened in Chinese cabbage protein sequence databases using TBtools. Combining these methods, duplicate and low-coverage sequences were removed, and protein sequences initially screened in the SwissProt database (National Center for Biotechnology Information, NCBI) were further compared. Candidate sequences were scrutinized based on their annotations and validated through one-by-one comparison with SMART (http://smart.emblheid-elberg.de/), HMMER (https://www.eb-i.ac.uk/Tools/hmm-er/) and Pfam (http://pfa-m.xfam.org/) databases. Finally, the *BrJMJ* genes were numbered based on their chromosomal locations.

### Phylogenetic tree establishment

To elucidate the *JmjC* family tree and functional features in Chinese cabbage, we gathered gene IDs and protein sequences of AtJMJs, OsJMJs, ZmJMJs, BpJMJs, and GmJMJs from published studies on *Arabidopsis* (At) [32], *Oryza sativa* (Os) [33], *Zea mays* L (Zm) [21], *Betula pendula* (Bp: silver birch) [34], and *Glycine max* (Gm, soybean) [7]. In total, 158 JmjC proteins from these species were compared and analyzed using MEGA 11.0. A root-less phylogenetic tree was constructed using the neighbor-joining method. The evolutionary tree was

embellished and clustered for annotation using Evolview (www.ev-olgenius.info/evol-view/). The evolutionary relationships between *JmjC* family genes in *Brassica rapa* and other species were analyzed.

## Gene structure, protein structural domain allocation, and cis-acting elements

Based on the annotated *Brassica rapa* genome files (GFF format), gene structures were analyzed and their structures were visualized using TBtools [35]. The protein structure of each BrJMJ-encoded protein was predicted using SMART (http://smart.embl-hei delberg.de/). Sequences 2,000 bp upstream of the BrJMJs transcriptional start site were extracted using using the 'GTF/GFF3 Sequences Extract'function of TBtools and submitted to the PlantCARE website (https://bioinfor-matics.psb.ugent.b-e/webtools/plantcare/html/) for prediction of potential cis-acting elements. The results were visualized using TBtools.

## Chromosomal distribution and gene duplication analysis

*JmjC* gene family members were localized to each chromosome based on genome annotations and were plotted for analysis using TBtools. Gene duplication analysis in Chinese cabbage was performed using BLAST comparisons with BrJMJ proteome sequences. Covariances among *JmjC* gene family members were visualized and plotted using MCScanX.

## Transcriptome analysis of *BrJMJ* gene family in different tissues

Transcriptomes data of *Brassica rapa* stems (SRX213893), siliques (SRX213892), roots (SRX213890), leaves (SRX213888), flowers (SRX213887), and callus tissues (SRX213886) were downloaded from the NCBI database (https://www.ncbi.nlm.ni-h.gov/sra). The TPM values of *BrJMJ*s in transcripts from different tissues were obtained by analysis using the RNA-seq tool in TBtools, and visual heatmaps were generated using the Heatmap tool in the software.

## Analysis of expression specificity in response of *BrJMJ* genes to abiotic stress by qRT–PCR

RNA was extracted from stress-treated Chinese cabbage samples (Cd, osmotic, salt, cold, and heat stress) using the MolPure®TRIeasy Plus Total RNA Kit (Yeasen Biotechnology, Shanghai, China). RNA quality was determined using an ultra–micro spectrophotometer (Thermo Electron Corp, USA), with an OD260/280 ratio of 1.8–2.0. The cDNA was synthesized using the Hifair® III 1$^{st}$ Strand cDNA Synthesis SuperMix for qRT-PCR (Yeasen). DNAMAN 6.0 was used to design the BrJMJ qRT-PCR primers.

The reaction mixture for qRT–PCR contained 1.0 μL cDNA, 10 μL Universal Blue SYBR Green Master Mix, 0.5 μL each of the forward and reverse primers, and 8.0 μL sterile ultrapure water. The procedure was as follows: pre-denaturation at 95°C for 2.0 minutes, denaturation at 95°C for 10 seconds, and annealing at 60°C for 30 seconds, with synthesis of 40 cycles. The primer sequences were listed in S1 Table (*BrActin7* was used as the internal reference gene), and target gene relative expression was calculated using the Eq $2^{-\Delta\Delta Ct}$ method.

## Statistical analysis

Three biological replicates were used for each stress treatment in qRT-PCR. Significant differences in gene expression were detected using Duncan's method in SPSS 22.0. TBtools and Origin were used to analyze and visualize plots.

## Results

### Identification and phylogenetic analyses of the *JmjC* family

Based on the *Brassica rapa* genome published in the Brassicaceae Database (BRAD) [21], we identified 29 *BrJMJs* (*BrJMJ1–29*) after eliminating false discovery and sequence redundancy (S1 Table). Using Mega 11, a phylogenetic tree was generated based on 158 JmjC protein sequences from six species, including Chinese cabbage, *Arabidopsis*, rice, maize, birch, and soybean (Fig 1). Based on the phylogenetic tree, the *JmjC* family can be split into five subfamilies, including the JMJD6, KDM3/JHDM2, KDM4/JHDM3, KDM5/JARID1, and JmjC-only domain subfamilies. In addition to the JmjC-only domain subfamily, the remaining four subfamilies all contain *BrJMJ* genes from Chinese cabbage, *Arabidopsis*, rice, maize, birch, and soybean. Indicating that these four subfamilies were retained among the six species [36]. KDM3/JHDM2 was the largest subfamily, with 48 homologous JmjC protein sequences; KDM5/JARID1 was the second largest, with 39 homologous JmjC protein sequences; and JMJD6 was the smallest, with 16 homologous JmjC protein sequences. In addition, KDM4/

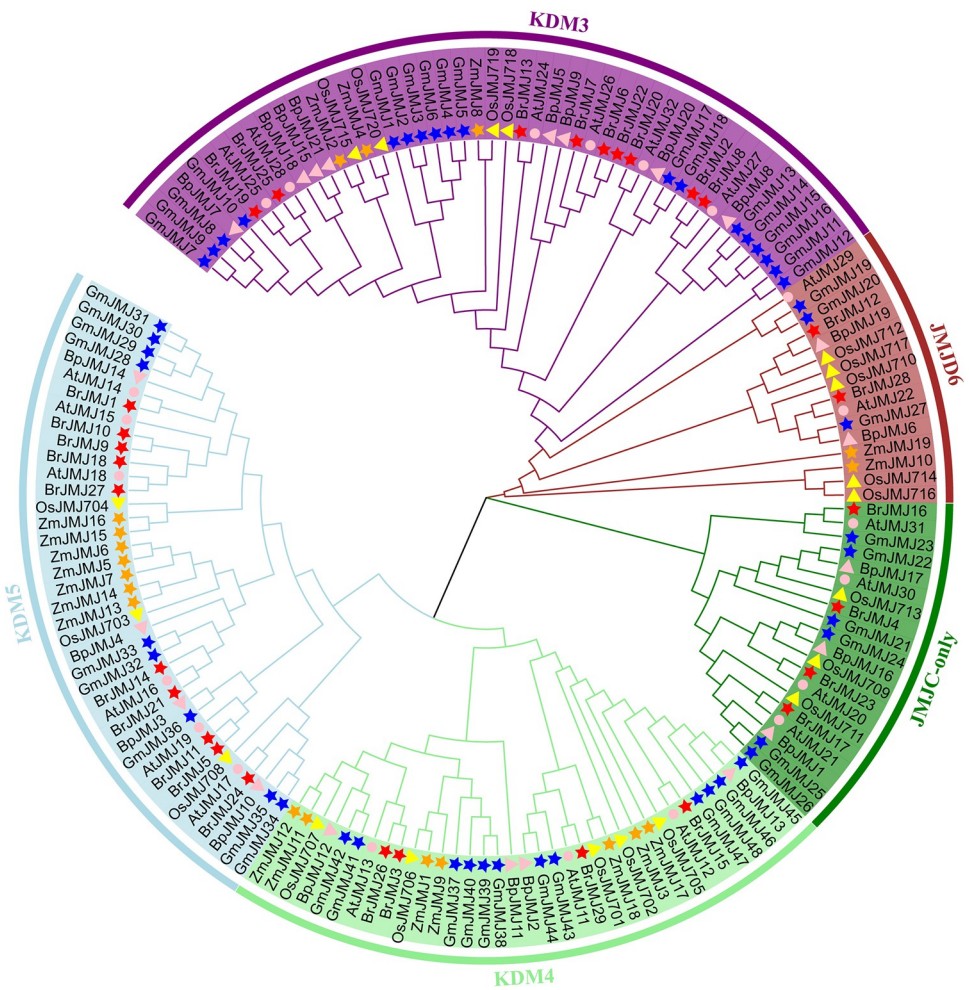

**Fig 1. Phylogenetic relationships of *JmjC* family members in *Brassica rapa* (Br), *Arabidopsis thaliana* (At), *Zea mays* (Zm), *Oryza sativa* (Os), *Betula platyphylla* (Bp), and *Glycine max* (Gm).** Using MEGA11.0, a phylogenetic tree was constructed for 158 JmjC protein sequences from the six species. Red stars, pink circles, orange stars, yellow triangles, pink triangles, and blue starts: Br, At, Zm, Os, Bp, and Gm, respectively.

JHDM3 and JmjC-only domains had 35 and 20 homologous JmjC protein sequences, respectively. The JmjC-only domain subfamily does not exist in maize.

The BrJMJs were closely associated with the AtJMJs, followed by OsJMJs, ZmJMJs, and GmJMJs. Among members of subfamilies, the BrJMJs were most evolutionarily distant from the BpJMJs, possibly reflecting the significant evolutionary gap between herbaceous and woody species. Over >55% of the BrJMJs showed the highest sequence similarity with *Arabidopsis* homologs. For example, in the KDM3 subfamily, BrJMJ25 of Chinese cabbage exhibited high homology with *Arabidopsis* AtJMJ28 (bootstrap values: 100%). In the KDM5 subfamily, Chinese cabbage BrJMJ27 and *Arabidopsis* AtJMJ18 (bootstrap values: 100%) had great evolutionary similarities, so they belong to homologous proteins. It is suggested that a high degree of sequence homology between the BrJMJs of Chinese cabbage and the AtJMJs of *Arabidopsis*, likely attributed to both species belonging to the crucifer family.

## Gene structure and conserved structural domains

The analysis of protein structural domains helps to reveal the functional and evolutionary relationships of the proteins, shedding light on potential similarities in their functions. To further investigate the differences and commonalities in the structural domains of the different BrJMJs proteins, we performed protein structural domain analysis using the SMART tool (Fig 2A). Our analysis showed that JmjC proteins contain 14 structural domains, with the JmjC structural domain being the most widely distributed, with the JmjC structural domain obviously present in all BrJMJ proteins. The JmjN structural domain was the second most widely distributed, present in 14 BrJMJ proteins. BrJMJ14 exhibited the greatest complexity in structural

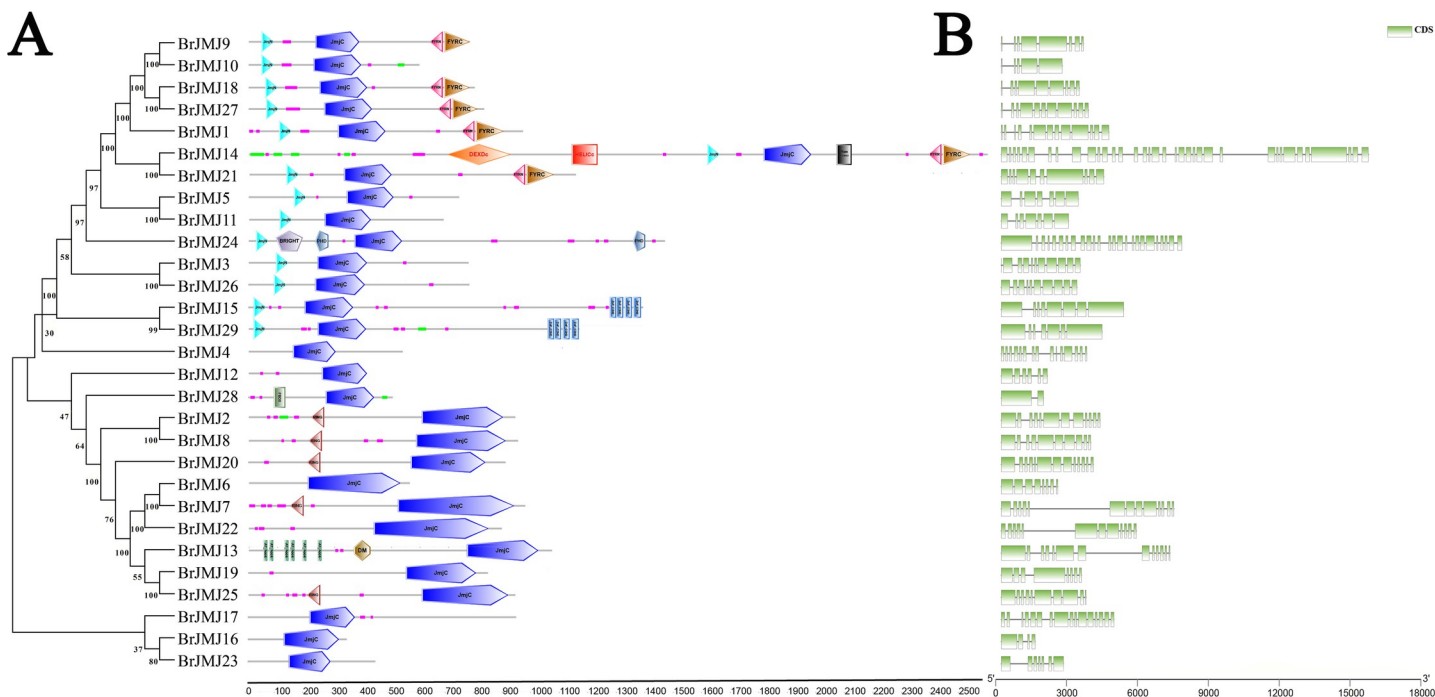

**Fig 2. Conserved structural domains of BrJMJ proteins in Chinese cabbage.** JmjC, Jumonji C domain; JmjN, Jumonji N domain; DEXDc, DEAD-like helicases superfamily; HELICc, Helicase superfamily c-terminal domain; PHD, plant homeobox domain; ARID, AT-rich interaction domain; FYRC, "FY-rich" domain C-terminal; FYRN, "FY-rich" domain N-terminal; F-box FBOX a receptor for ubiquitination targets; ZnF_C2H2, zinc-finger of C2H2-type; AT_hook, DNA binding domain with preference for A/T rich regions; DM, Dsx and Mab-3; RING, really interesting new gene (A). Gene structure analysis of BrJMJs in Chinese cabbage, green boxes, exons; black lines, introns. The sizes of exons and introns can be estimated using the scale at the bottom (B).

domains, having six structural domains: JmjC (Jumonji C), JmjN (Jumonji N), DEXDc (DEAD-like helicases superfamily), HELICc (Helicase superfamily c-terminal domain), FYRN ("FY-rich" domain N-terminal) and FYRC ("FY-rich" domain C-terminal) [37]. Among them, the FYRN and FYRC structural domains may promote the function of JmjC by interacting with other proteins [7]. BrJMJ24 contained the specific ARID (AT-rich interaction domain) [7] and PHD (Plant Homeodomain) structural domains. Among them, the PHD structural domain recognized as a versatile epigenetic reader for recognizing H3K4me3, H3K4me, or H3K14ac [38], so we speculate a potential role for BrJMJ24 in versatile epigenetic. BrJMJ15 and BrJMJ29 contained the specific ZnF_C2H2 (zinc-finger of C2H2-type) domain [7]. BrJMJ28 contained a structural F-box (FBOX a receptor for ubiquitination targets) [14], and BrJMJ13 had two specific protein structural domains, DM (Dsx and Mab-3) and AT-hook (DNA binding domain with preference for A/T rich regions) [37]. Additionally, five proteins (BrJMJ2, BrJMJ7, BrJMJ8, BrJMJ20 and BrJMJ25) contained RING (really interesting new gene) [7] as a specific structure. Different BrJMJs have several conserved and specialized domains, which means that the protein structure of the *BrJMJ* family maybe diverse.

The intron and exon structure are an important clue to understand the gene evolutionary relationship and functional diversification within a gene family. To further analyze the exon-intron structure and conserved structural regions of the *JmjC* gene family members, we mapped the structures of the 29 *BrJMJs* using TBtools. Members of the *JmjC* gene family contained 2.0–36 exons, with the majority having between 7.0 to 16 exons (Fig 2B). Notably, *BrJMJ28* contained the fewest exons (two), while *BrJMJ14* the most (36). The large variation in the number of exons may reflect the diversity of *JmjC* gene family members in Chinese cabbage. Furthermore, we observed a high level of consistency in gene structure among specific groups of genes, such as *BrJMJ20/ BrJMJ2/ BrJMJ8*, *BrJMJ5/ BrJMJ11*, *BrJMJ19/ BrJMJ25*, *BrJMJ15/ BrJMJ29*, and *BrJMJ3/ BrJMJ26*, which have the same exon/intron arrangement and number as well as very similar exon lengths.

Conserved structural domains of BrJMJ proteins in Chinese cabbage. JmjC, Jumonji C domain; JmjN, Jumonji N domain; DEXDc, DEAD-like helicases superfamily; HELICc, Helicase superfamily c-terminal domain; PHD, plant homeobox domain; ARID, AT-rich interaction domain; FYRC, "FY-rich" domain C-terminal; FYRN, "FY-rich" domain N-terminal; F-box FBOX a receptor for ubiquitination targets; ZnF_C2H2, zinc-finger of C2H2-type; AT_hook, DNA binding domain with preference for A/T rich regions; DM, Dsx and Mab-3; RING, really interesting new gene (A). Gene structure analysis of BrJMJs in Chinese cabbage, green boxes, exons; black lines, introns. The sizes of exons and introns can be estimated using the scale at the bottom (B).

## Cis-acting element analysis

To further elucidate *BrJMJs* regulation in response to biotic and abiotic stress, we used PlantCARE to analyze the promoter cis-acting elements within the 2000 bp sequence upstream of the start codons of each *BrJMJ* gene in the Chinese cabbage (Fig 3). Our analysis revealed a variety of phytohormone response elements within the JmjC family, including methyl jasmonate (in 37.89% of the genes), abscisic acid (29.52%), gibberellin (12.78%), growth hormones (11.45%) and salicylic acid (8.37%) elements. Additionally, the *JmjC* family contains many cis-acting elements associated with defense or responses to abiotic stresses, such as light-responsive element (73.98%) and anaerobically induced (16.06%). The light-responsive element was present in all 29 genes and anaerobic stress-related cis-acting elements were present in most of the genes (86.21%). Growth and biological process responsive elements included meristem

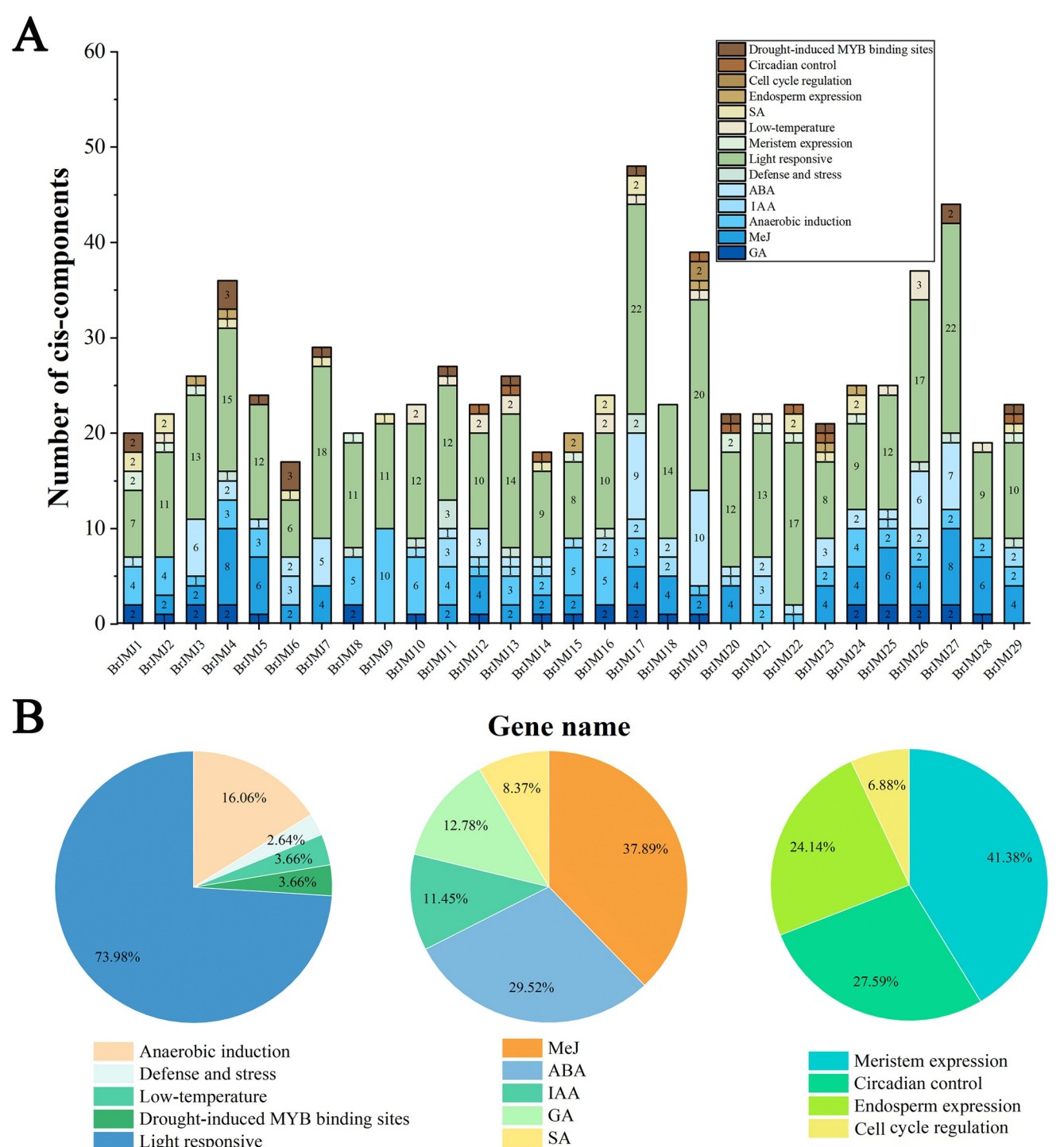

**Fig 3. Cis-acting elements of *BrJMJ*s in *Brassica rapa*.** Different color blocks represent different components. Relative frequencies in *BrJMJ*s of promoter cis-regulatory elements (CREs), including environment and stress-responsive CREs, phytohormone response CREs, and growth and biological process-responsive CREs.

expression-responsive elements (41.38%), physiological rhythm-regulating action elements (27.59%) and endosperm expression-specific elements (24.14%).

In terms of gene regulation, the promoters of *BrJMJ17*, *BrJMJ19*, and *BrJMJ27* were found to contain a variety of light-oriented response elements and ABA response elements. *BrJMJ4*, *BrJMJ27*, and *BrJMJ28* were identified to have multiple elements associated with the methyl jasmonate response. Additionally, *BrJMJ9* contained numerous elements associated with anaerobic responses, suggesting that this gene may play an important role in an anoxic environment. *BrJMJ19* was the only member that contained cis-acting elements related to cell-cycle responses, whereas *BrJMJ17* had the most cis-acting elements, including those related to light-response and low-temperature, defense mechanism, anaerobic induction and phytohormone response. These findings suggest that the *BrJMJ* promoters contain numerous and diverse cis-acting elements, potentially acting across multiple abiotic stresses.

## Chromosomal distribution and gene duplication

Based on the Chinese cabbage genome annotation file and *JmjC* gene family member data, a chromosome localization map of *BrJMJ* family genes was constructed using TBtools (Fig 4A). The 29 *BrJMJ*s were found to be distributed across ten chromosomes, with chromosomes A02 and A07 each contained only one *BrJMJ*. Notably, 37.93% of the genes were distributed on chromosomes A03 and A09, indicating an uneven distribution of the 29 *BrJMJ*s on the chromosomes. It is suggested that *BrJMJ* genes might have undergone duplication or loss in the process of evolution.

Three types of gene duplication events have been reported, including tandem duplication, segmental duplication, and whole-genome duplication [39]. To investigate duplication of *BrJMJ*s on chromosomal segments, covariance analysis of the *BrJMJ* family genes was conducted by using MCScanX. This analysis revealed that 10 pairs of *JmjC* genes exhibited duplication events among the 29 *BrJMJ* genes (Fig 4B). Interestingly, the 10 pairs of duplicated genes were found to be located on different chromosomes, suggesting that they were produced by segmental duplication, such as *BrJMJ5/ BrJMJ11*, *BrJMJ13/ BrJMJ26*, and *BrJMJ18/ BrJMJ27*. Therefore, this indicates that gene segmental fragment duplication is likely the main method of expansion of the *JmjC* gene family in Chinese cabbage.

## Profiling of tissue-specific *BrJMJ* gene expression

Based on the analysis of transcriptome data from the Brassicaceae Database, Fig 5 displayed a heatmap of tissue-specific *BrJMJ* expression in various tissues including callus, flower, leaf, root, silique, and stem. The majority of the 29 *BrJMJ*s exhibited high expression levels in the reproductive organs such as siliques and flowers. Some genes (*BrJMJ14/ BrJMJ19/ BrJMJ21/ BrJMJ27*) expression was high in the flowers, while other genes (*BrJMJ8/ BrJMJ9/ BrJMJ10*) displayed high levels of expression in the siliques, suggesting a potential role of histone demethylation in the growth of reproductive organs in Chinese cabbage. In addition, approximately one-third of *BrJMJ*s were highly expressed in stems and root, with some genes (*BrJMJ7/ BrJMJ16/ BrJMJ29*) being expressed in callus tissues. Interestingly, *BrJMJ14/ BrJMJ21* belonged to the same branch of the phylogenetic tree (Fig 1A), but showed different expression patterns in the same tissues. As a result, it can be speculated that they acquire different functions after the duplication event.

## *BrJMJ* expression under Cd stress

To further investigate the functions of *BrJMJ*s under Cd stress, the relative expression of the 29 *BrJMJ*s was quantified at Cd concentrations of 2.0, 4.0, 6.0, 8.0 and 10 mg/L by qRT–PCR. A

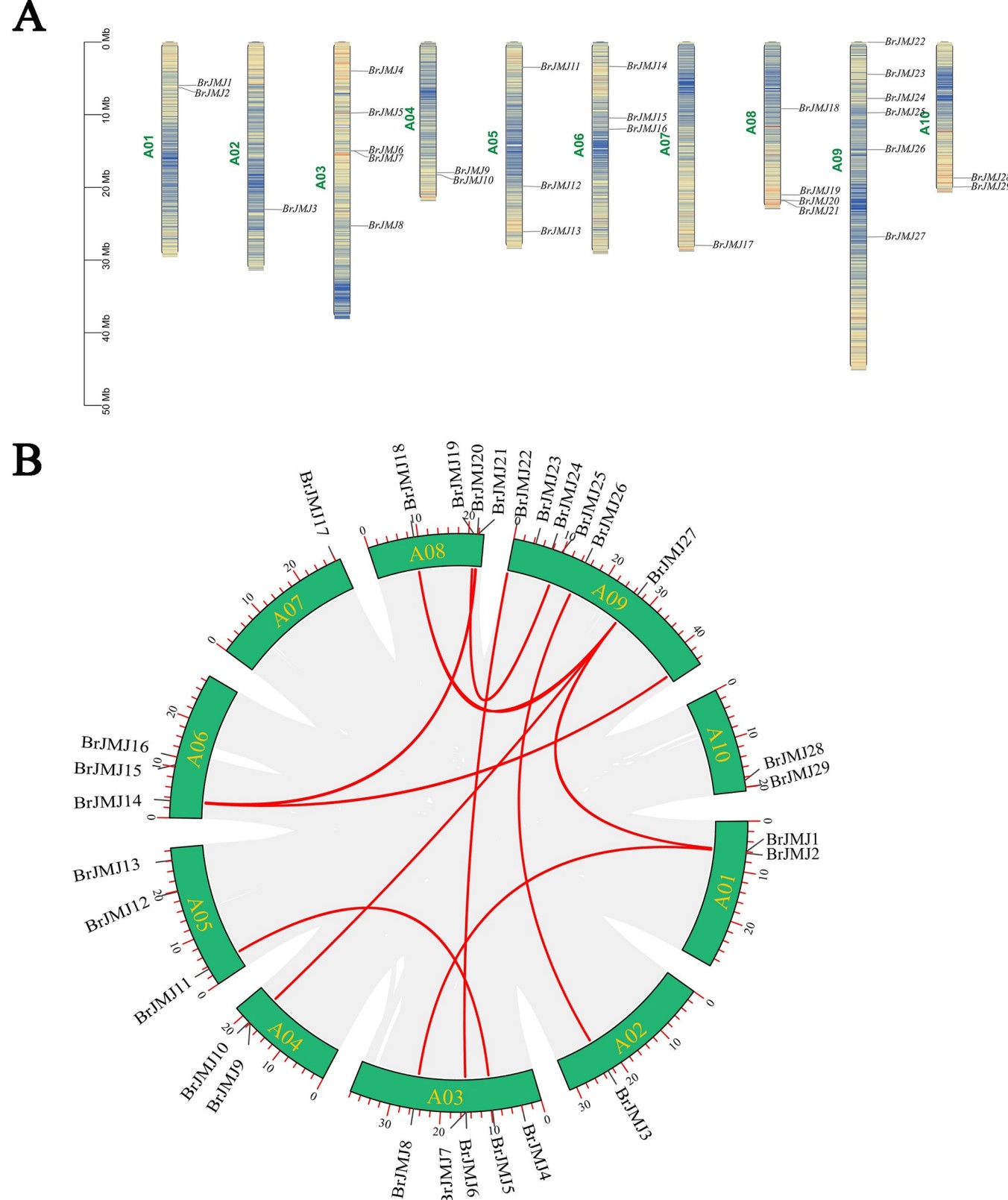

**Fig 4.** Chromosome distribution (A) and schematic representation of inter-chromosomal relationships (B) of the *BrJMJs* in *Brassica rapa*. Repeated *BrJMJ* pairs are highlighted with red lines.

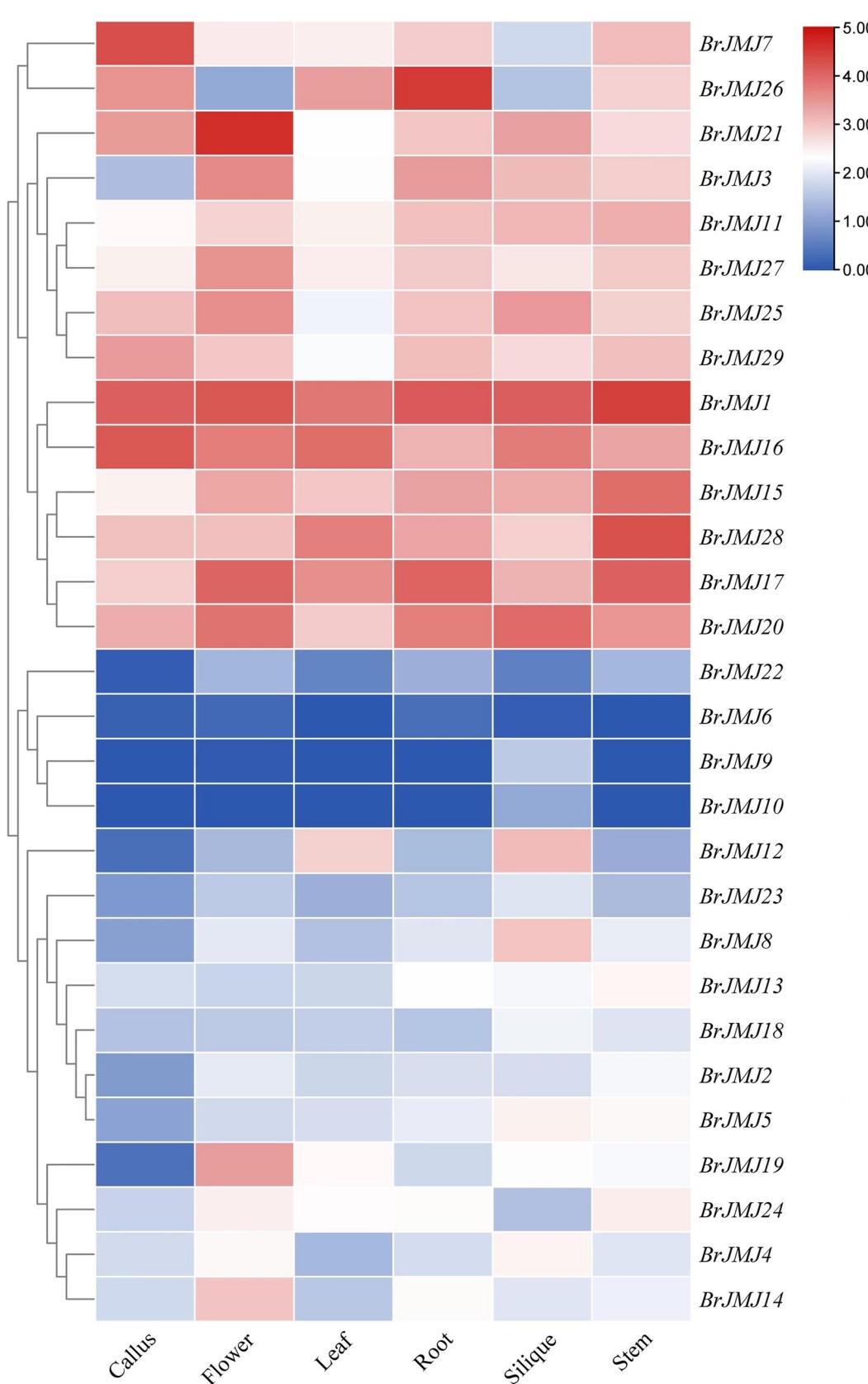

**Fig 5. Tissue-specific _BrJMJ_ expression.** The heatmap was constructed using TBtools, based on the fragments per kilobase of transcripts per million mapped reads (FPKM) values of _BrJMJ_s in the tissue-specific transcriptome data.

heatmap was generated to visualize the expression patterns (Fig 6A–6C). With the exception of _BrJMJ10_, the remaining 28 _BrJMJ_s were differentially expressed under Cd treatment, with over >52.0% showing upregulated expression (Fig 6A). Under the same treatment time, the higher the cadmium concentration, the smaller the leaves of Chinese cabbage (Fig 6B). The expression of certain genes (_BrJMJ1_, _BrJMJ2_, _BrJMJ5_, _BrJMJ13_, _BrJMJ21_ and _BrJMJ24_) increased gradually with increasing Cd concentrations and their expression levels were positively correlated (Fig 6C). Additionally, _BrJMJ3_, _BrJMJ7_, _BrJMJ15_, _BrJMJ18_, _BrJMJ20_, _BrJMJ25_, _BrJMJ27_, and _BrJMJ29_ were significantly upregulated ($p<0.05$) at high Cd concentrations (Fig 6C). _BrJMJ15_, _BrJMJ20_, _BrJMJ25_, _BrJMJ27_, and _BrJMJ29_ were significantly upregulated at a Cd concentration of 8.0 mg/L, with a lesser upregulation at 10 mg/L. The expression of _BrJMJ11_/ _BrJMJ18_ was significantly upregulated ($p<0.05$) at a Cd concentration of 6.0 mg/ L, but showed less upregulation at Cd levels ≥8.0 mg/L (Fig 6C). _BrJMJ9_, _BrJMJ16_, _BrJMJ19_, _BrJMJ22_, and _BrJMJ23_ exhibited downregulation with Cd treatment. _BrJMJ19_ expression declined gradually with increasing Cd content (Fig 6C). This suggests that numerous _BrJMJ_s may play a role in resistance to Cd stress.

## _BrJMJ_ expression under other abiotic stresses

The expression of JmjC proteins under environmental stimulation is specific [40]. Using qRT-PCR, we investigated _JmjC_ family member expression in the leaves under osmotic, salt, cold, and heat stress. A heatmap was generated using the TBtools (Fig 7A). Among the 29 _BrJMJ_s analyzed, the remaining _BrJMJ_s showed differential expression under these abiotic stresses, except for _BrJMJ10_/ _BrJMJ12_/ _BrJMJ23_.

Under osmotic stress, 65.4% of the _BrJMJ_s were upregulated (log2FC>1.5) (Fig 7A). And the expression levels of _BrJMJ2_ was more than 2.0-fold higher with expression levels in the control group following 6.0 hours of osmotic treatment. Under salt stress, 50.0% were upregulated, with expression levels correlating positively with stress treatment duration (log2FC>1.5) (Fig 7A). However, downregulation of _BrJMJ18_ expression increased with treatment time. _BrJMJ20_ was upregulated but upregulation of its expression tended to be stable over treatment time. The expression of _BrJMJ26_ was unaffected by salt stress (Fig 7B). Under cold stress, 46.2% of _BrJMJ_ genes were downregulated, with the degree of downregulated expression increasing over treatment time (log2FC>1.5) (Fig 7A). Under heat stress, 74.0% of _BrJMJ_ genes were upregulated (log2FC>1.5). It can be speculated that the gene expression of the _BrJMJ_ family is significantly affected by low-temperature stress. However, some genes (_BrJMJ13_/ _BrJMJ14_/ _BrJMJ21_/ _BrJMJ26_) expressions reached their highest levels after 3.0 hours of cold stress but were subsequently inhibited by prolonged exposure to cold stress. Notably, _BrJMJ29_/ _BrJMJ5_/ _BrJMJ2_ were upregulated under both heat and cold stress.

Combining the four abiotic stresses (cold, heat, osmotic, and salt), it was observed that the expression of _BrJMJ1_/ _BrJMJ15_/ _BrJMJ25_ was generally unaffected by the four types of abiotic stress, except for slight upregulation under 9.0 hours salt treatment. Conversely, _BrJMJ2_, _BrJMJ4_, _BrJMJ5_, and _BrJMJ29_ were upregulated under all stress treatments. In particular, the expression level of _BrJMJ29_ was upregulated under the four abiotic stresses compared with the control group, and the expression was positively correlated with stress-treatment duration (Fig 7B). These findings indicate that _BrJMJ_s may have significant roles in the responses of Chinese cabbage to abiotic stresses.

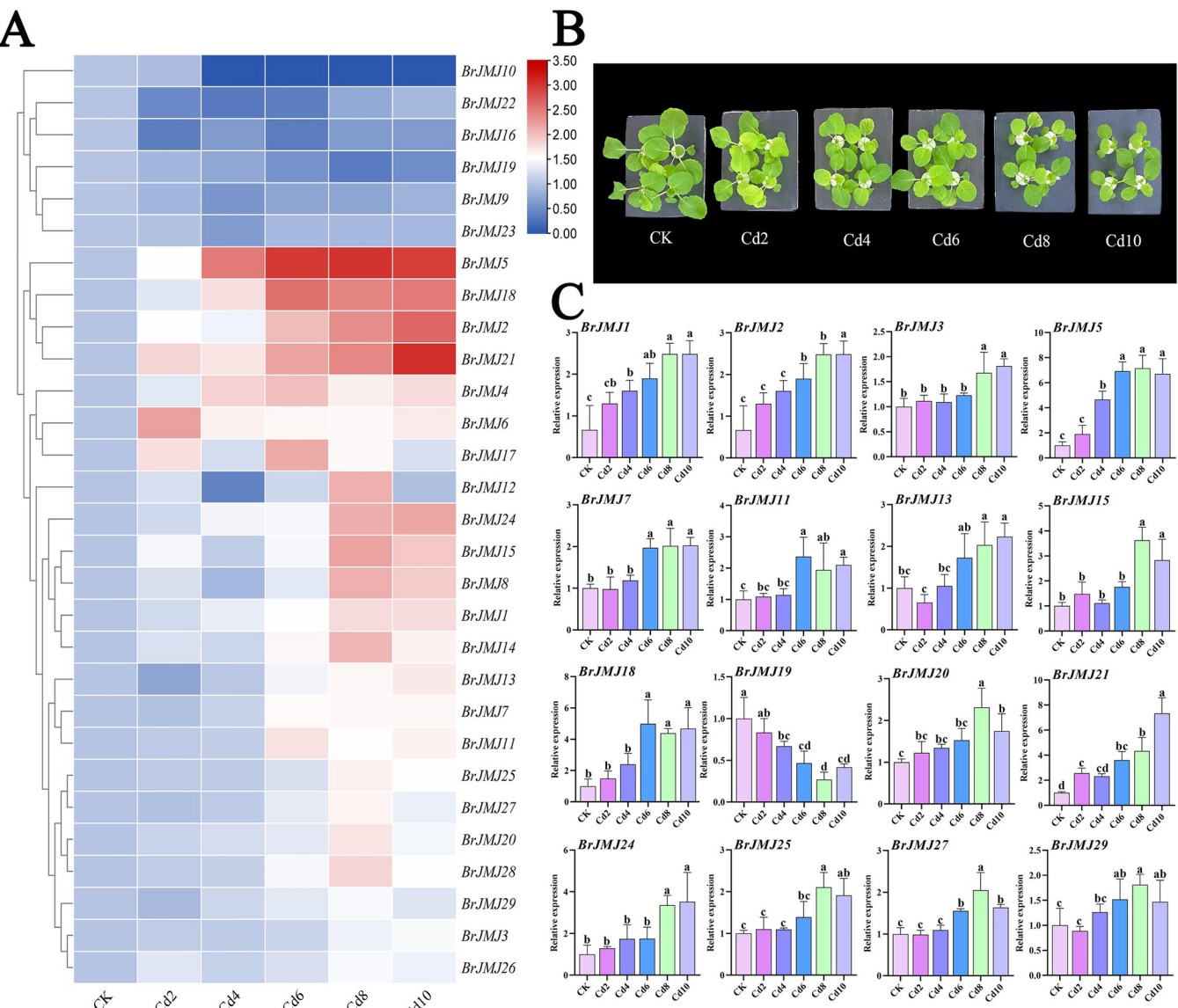

**Fig 6. Profiling of *BrJMJ* expression under Cd stress.** Variation in *BrJMJ* gene expression with Cd stress, via qRT–PCR, with *BrACTIN7* as an internal reference gene. *BrJMJ* relative expression was calculated using the $2^{-\Delta\Delta Ct}$ method. The results were visualized as a heatmap using TBtools (A). Growth status of Chinese cabbage under Cd stress. Scale bar, 4 cm (B). Using the same RT-qPCR results as in Fig 6A, a histogram of representative genes were generated to analyze significance. *Brassica rapa* growth varied significantly under Cd stress, with different lowercase letters indicating significant differences (p<0.05). The *x* axis represents different concentrations of Cd stress treatment, and the *y* axis represents relative expression (C).

## Discussion

Epigenetic regulation has garnered increasing attention in the realms of plant growth, development, and stress responses [21]. Within the realm of epigenetic gene expression regulation, PMTs and demethylases are pivotal in modulating histone methylation status [41]. Additionally, JmjC proteins play a crucial role in regulating epigenetic processes as well as the growth and development of plants [7]. The *JmjC* family has been extensively studied in various species such as *Arabidopsis* [32], rice [33], birch [34], soybean [7], maize [21], *Pyrus bretchneideri* [41], and Upland cotton [37]. This research aimed to identify and characterize the expression of *JmjC* family members in Chinese cabbage. The study delved into the gene structure, conserved

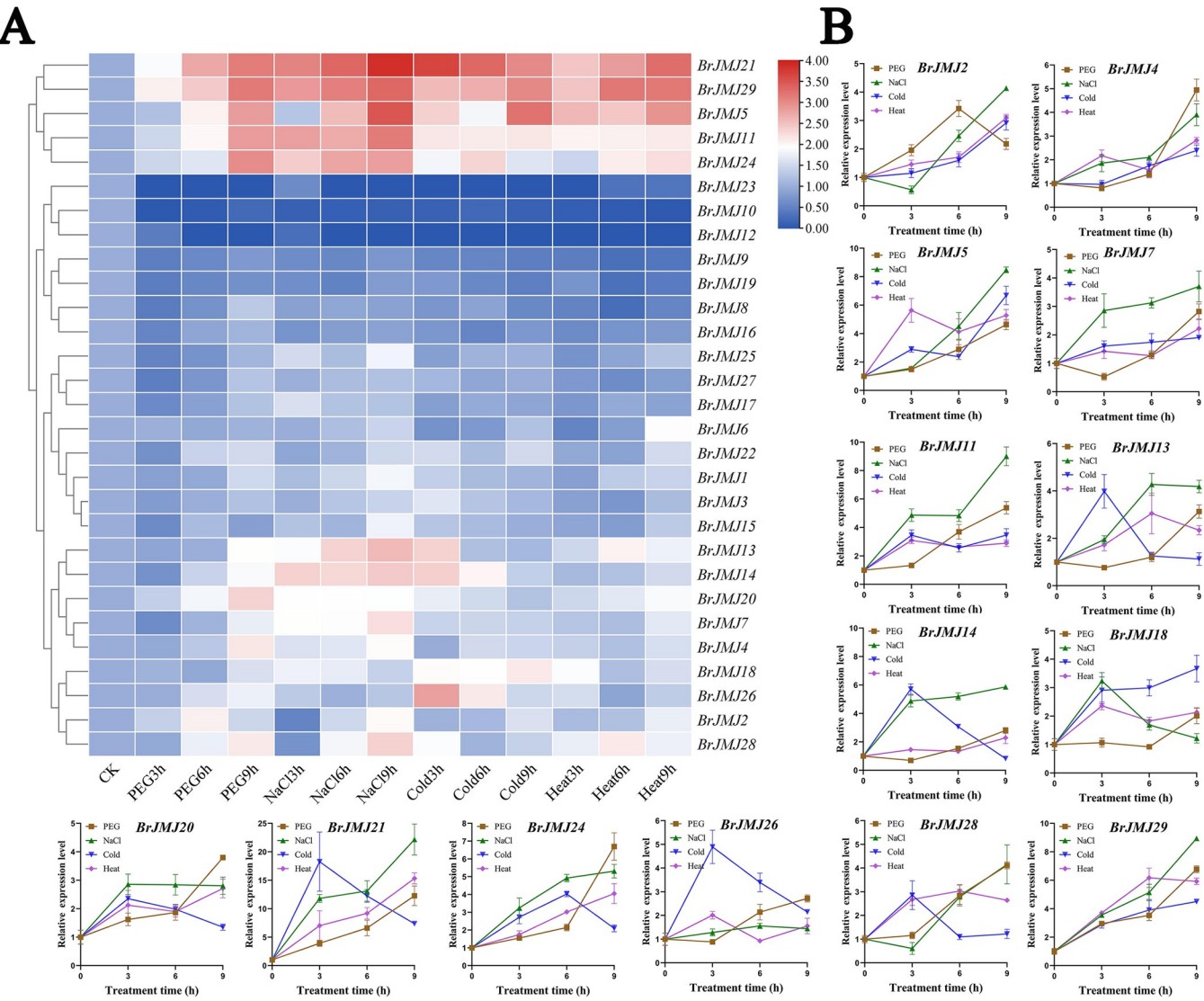

**Fig 7. Profiling of *BrJMJ* expression under other abiotic stresses.** Expression under drought, salt, cold, and heat stress of the 29 *BrJMJ*s, via qRT–PCR. *BrACTIN7* was an internal reference gene. *BrJMJ* relative expression was calculated using the $2^{-\Delta\Delta Ct}$ method. The results were visualized as a heatmap using TBtools (A). The significant changes of representative genes under four abiotic stresses are indicated by the length of vertical lines ($p<0.05$). Red polyline, green polyline, blue polyline and purple polyline are respectively represented Expression under osmotic, salt, cold, and heat stress of the 29 *BrJMJ*s. The *x* axis represents the different treatment times of the four abiotic stresses (osmotic, salt, cold, and heat stress), and the *y* axis represents the relative expression (B).

structural domains, cis-acting elements, chromosomal distribution, and gene duplication of *BrJMJ*s using specialized software.

The number of *JmjC* family members varies among species, such as in *Arabidopsis* (21 *JmjC* genes) [32], rice (20 *JmjC* genes) [33], maize (19 *JmjC* genes) [21], and birch (21 *JmjC* genes) [34], while significantly more *JmjC* genes were identified in soybean (48 *JmjC* genes) [7]. This indicates potential differences in evolutionary history among species. In this study, phylogenetic analysis identified 29 *JmjC* genes in Chinese cabbage, serving as a valuable reference for genetic evolution studies in this species probably. The *BrJMJ*s were classified into five subfamilies (JMJD6, KDM3/JHDM2, KDM4/JHDM3, KDM5/JARID1, and JmjC-only domain) in Chinese cabbage, which is consistent with findings in other plant species like *Arabidopsis*,

soybean, birch, rice and *Lycopersicon esculentum* [42]. This suggests potential functional similarities between *BrJMJ*s and *JmjC* genes across these species. The majority of *BrJMJ*s clustered closely with *AtJMJ*s on small branches, likely due to both species being *Brassicaceae* (cruciferae) dicotyledonous plants, distinct from monocotyledonous plants like rice and maize.

In plants, tandem and segmental duplication are the most common causes of gene family expansion [43,44], with segmental duplication occurring frequently in slower-evolving gene families [45]. Gene duplication leads to functional diversity and the generation of new genes, with significant implications for environmental adaptation, biological evolution, and continued species evolution probably [46]. Among the 29 *BrJMJ*s found in Chinese cabbage, 10 pairs are the result of segmental duplication. We speculate that segmental duplication may have contributed to expansion of *JmjC* gene family in Chinese cabbage, potentially playing a crucial role in environmental adaptation.

Cis-acting promoter elements interact with transcription factors to regulate gene transcription [47–49]. The function of the *JmjC* family members can be inferred from the analysis of cis-acting elements [46]. In this study, *BrJMJ17* had the most cis-acting elements, including those related to light-response, low-temperature, defense mechanism, anaerobic induction and phytohormone response. It can be speculated that *BrJMJ17* may play an important role in plant stress. Interestingly, *JMJ524* responds to circadian rhythms and was upregulated by GA treatment in tomato [50]. However, this phenomenon has not been found in Chinese cabbage, which requires further validation. These results reveal that *JmjC* genes potentially regulate plant metabolic processes by involving biotic and abiotic responses.

*JmjC* gene expression varies among species and tissue types. And *JmjC* gene demethylases play an important role in plant growth and development [51]. In wheat, *JmjC* expression was higher in nutritional tissue growth than in reproductive organ growth and was higher in roots and spikes than in leaves and seeds [46]. In Chinese rose, most of the *JmjC* genes were highly expressed in reproductive tissues but not in nutritional tissues [14]. Here, most of the *BrJMJ*s were highly expressed in the reproductive organs (siliques and flowers) and some were highly expressed in the stems and roots. These results indicate that histone demethylation plays a potential role in both nutritional tissue growth and reproductive organ growth of Chinese cabbage, particularly during reproductive organ growth.

*BrJMJ14/ BrJMJ21* belonged to the same branch of the phylogenetic tree (Fig 1), but showed different expression patterns in the same tissues. It can be speculated that they may have acquired different functions following the duplication event. Comparable expression patterns of *JmjC* genes have also been observed in cotton [37] and soybean [7].

When plants are subjected to heavy metal stress in the environment, their organs repress or initiate certain gene expression by altering DNA methylation in response to achieve resistance to heavy metal stress [24]. In *Arabidopsis*, Cd can induce an increase in DNA methylation [52]. In wheat (*Triticum aestivum*), Cd phytotoxicity can alter DNA methylation levels to confer heavy metal tolerance [53]. Additionally, overexpression of the histone demethylase gene *SlJMJ524* from tomato enhances Cd tolerance in *Arabidopsis* by regulating metal transporter-related protein genes and flavonoid content [54]. Over >52.0% of *BrJMJ*s were differentially upregulated in expression level under Cd stress. Among them, the expression levels of *BrJMJ1/ BrJMJ2/ BrJMJ5/ BrJMJ13/ BrJMJ21/ BrJMJ24* gradually increased with Cd concentrations increasing and had a positive correlation. These findings highlight the significance of DNA methylation in Chinese cabbage resistance to Cd toxicity [55].

Different abiotic stresses can regulate the expression of *JmjC* genes, thus influencing plant growth and development [34]. Our study observed differential expression of most *BrJMJ*s under four abiotic stresses (osmotic, salt, cold, and heat). In particular, over >74.0% of *BrJMJ*s were upregulated in expression level under heat stress, such as *BrJMJ2/ BrJMJ21/ BrJMJ24/*

*BrJMJ29*. H3K36me2/3 demethylase can improve the resilience of Chinese cabbage under heat stress [56]. We speculate that these genes may play an important regulatory role in improving Chinese cabbage resistance to high temperature stress. Similarly, in maize, the expression of seven genes of the *ZmJMJs* family (*ZmJMJ3*, *ZmJMJ5*, *ZmJMJ8*, *ZmJMJ10* and *ZmJMJ19*) can be upregulated with heat stress [21]. Our findings revealed that half of *BrJMJ*s were significantly downregulated under cold stress (log2FC>1.5), implying a response to external low temperatures by downregulating *BrJMJ*s in Chinese cabbage probably, which is consistent with the findings in birch [34]. Conversely, most *JmjC* gene members of the allotetraploid cotton species showed upregulate in response to cold stress [37]. *BrJMJ11*/ *BrJMJ21*/ *BrJMJ28*/ *BrJMJ29* were significantly upregulated under osmotic stress and salt stress. Similar findings were observed in cotton, where *GhJMJ40* and *GhJMJ34* showed upregulated expression [37], suggesting that these genes might have the same regulatory mechanisms under osmotic stress and salt stress in both cotton and Chinese cabbage. A similar phenomenon exists in *Jatropha curcas* L. (*Jatropha*), with *JcJMJ18* expression significantly upregulated under salt stress [15]. This suggests that *JmjC* family genes may actively respond to osmotic and salt stress by up-regulating their expression [15]. In conclusion, the varied expression patterns of *BrJMJ*s under different abiotic stress conditions imply their responsiveness to stress treatments.

## Conclusions

This study conducted a comprehensive analysis on the structural characteristics, evolutionary relationships, and gene expression of *JmjC* gene family members in Chinese cabbage. A total of 29 members were identified, showcasing a high level of conservation at the genome level. By analyzing the evolutionary relationships of *BrJMJ*s, a high degree of sequence homology between the BrJMJs of Chinese cabbage and the AtJMJs of *Arabidopsis* was observed. This may be attributed to their shared membership in the crucifer family. For these genes, the promoter cis-acting elements were enriched in response to stress related to light, low-temperature, anaerobic conditions and phytohormone treatment. The high expression of *BrJMJ*s in the siliques and flowers indicated a potential role of histone demethylation in reproductive organ growth. Furthermore, stress associated with Cd exposure, osmotic, salinity, cold, and heat induced their expression. These findings may provide a theoretical basis and reference for further research on the potential functions of Chinese cabbage *BrJMJ*s and improving abiotic stress tolerance in Chinese cabbage.

## Supporting information

**S1 Table. Detailed information on *BrJMJs* in Chinese cabbage.** It includes the gene name, gene ID, forward primer sequence, reverse primer sequence, and the gene name and gene ID of the closest *Arabidopsis thaliana* homolog for each of the *BrJMJ* genes.
(PDF)

## Author Contributions

**Data curation:** Xufeng Xiao, Ming Zhang.

**Formal analysis:** Fengrui Yin, Yuanfeng Hu, Xiaoqun Cao, Yan Xiang, Liangdeng Wang.

**Funding acquisition:** Xufeng Xiao, Ming Zhang.

**Investigation:** Yuanfeng Hu, Xufeng Xiao.

**Methodology:** Fengrui Yin, Yuanfeng Hu, Xiaoqun Cao, Yan Xiang, Liangdeng Wang, Yuekeng Yao, Meilan Sui.

**Resources:** Xufeng Xiao, Ming Zhang.

**Software:** Ming Zhang.

**Writing – original draft:** Fengrui Yin, Xufeng Xiao, Wenling Shi.

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
