## [Decision Letter · Decision Letter 0]

27 Sep 2024

PONE-D-24-35698JmjC domain-containing histone demethylase gene family in Chinese cabbage: genome-wide identification and expressional profilingPLOS ONE

Dear Dr. xu-feng,

Thank you for submitting your manuscript to PLOS ONE. After careful consideration, we feel that it has merit but does not fully meet PLOS ONE’s publication criteria as it currently stands. Therefore, we invite you to submit a revised version of the manuscript that addresses the points raised during the review process.

We look forward to receiving your revised manuscript.

Kind regards,

Steven G. Gray

Academic Editor

PLOS ONE

Journal requirements: 1. When submitting your revision, we need you to address these additional requirements. Please ensure that your manuscript meets PLOS ONE's style requirements, including those for file naming. The PLOS ONE style templates can be found at https://journals.plos.org/plosone/s/file?id=wjVg/PLOSOne_formatting_sample_main_body.pdf and https://journals.plos.org/plosone/s/file?id=ba62/PLOSOne_formatting_sample_title_authors_affiliations.pdf 2. We note that the grant information you provided in the ‘Funding Information’ and ‘Financial Disclosure’ sections do not match.  When you resubmit, please ensure that you provide the correct grant numbers for the awards you received for your study in the ‘Funding Information’ section. 3. Thank you for stating the following financial disclosure:  [The Natural Science Foundation of China (31860560) and the Natural Science Foundation of Jiangxi Province (20224BAB205027).].  Please state what role the funders took in the study.  If the funders had no role, please state: ""The funders had no role in study design, data collection and analysis, decision to publish, or preparation of the manuscript."" If this statement is not correct you must amend it as needed. Please include this amended Role of Funder statement in your cover letter; we will change the online submission form on your behalf. 4. Please amend the manuscript submission data (via Edit Submission) to include author Yan Xiang and Liangdeng Wang. 5. Please include captions for your Supporting Information files at the end of your manuscript, and update any in-text citations to match accordingly. Please see our Supporting Information guidelines for more information: http://journals.plos.org/plosone/s/supporting-information.  6. We notice that your supplementary table is uploaded with the file type 'other'. Please amend the file type to 'Supporting Information'. Please ensure that each Supporting Information file has a legend listed in the manuscript after the references list.

Additional Editor Comments:

The reviewer has suggested a minor revision

Reviewers' comments:

Reviewer's Responses to Questions

**Comments to the Author**

1. Is the manuscript technically sound, and do the data support the conclusions?

Reviewer #1: Yes

2. Has the statistical analysis been performed appropriately and rigorously? 

Reviewer #1: Yes

3. Have the authors made all data underlying the findings in their manuscript fully available?

Reviewer #1: Yes

4. Is the manuscript presented in an intelligible fashion and written in standard English?

Reviewer #1: Yes

5. Review Comments to the Author

Reviewer #1: I am pleased to see that the manuscript has been improved and my concerns have been addressed in the author's response. The article now presents a more sound piece of work, although I still have some reservations about the overall quality of the analysis.

Minor comment:

The authors should include in Table S1 the closest Arabidopsis thaliana homolog for each of the B. rapa genes. This addition would help the average reader identify putative homologs and potential functions of the B. rapa genes more easily.

6. PLOS authors have the option to publish the peer review history of their article (what does this mean?). If published, this will include your full peer review and any attached files.

Reviewer #1: No

---

## [Author Response · Author response to Decision Letter 0]

9 Oct 2024

Responds to the Editor’ comments: 

1.When submitting your revision, we need you to address these additional requirements. Please ensure that your manuscript meets PLOS ONE's style requirements, including those for file naming.

Response: Thank you very much for your professional advice. According to the format required by the journal, the size of the full text title, author byline (Page 1, line 5-6, line 14-17), affiliations (Page 1, line 8-12), the format of the references (Page 29-33, line 562-760), Table and Figure captions, file naming for Figures and Tables. Add each graphic heading as required after the paragraph in which they are first referenced. Figures are uploaded separately as individual files. The Acknowledgments (Page 27, line 539-541), Author Contributions (Page 27-28, line 542-552), and Supporting information (Page 33, line 761-764) are added at the end of the manuscript.

2.We note that the grant information you provided in the ‘Funding Information’ and ‘Financial Disclosure’ sections do not match. When you resubmit, please ensure that you provide the correct grant numbers for the awards you received for your study in the ‘Funding Information’ section.

Response: We feel sorry for our carelessness. In our resubmitted manuscript, we have carefully checked and ensured that we provide the consistent and correct grant numbers in both the ‘Funding Information’ and ‘Financial Disclosure’ sections.

3.Thank you for stating the following financial disclosure: [The Natural Science Foundation of China (31860560) and the Natural Science Foundation of Jiangxi Province (20224BAB205027)]. Please state what role the funders took in the study. If the funders had no role, please state: ""The funders had no role in study design, data collection and analysis, decision to publish, or preparation of the manuscript."" If this statement is not correct you must amend it as needed. Please include this amended Role of Funder statement in your cover letter; we will change the online submission form on your behalf.

Response: Thank you for your help. We have additionally added the funder's role in this study both in the “Funding” section of the revision manuscript and the cover letter. Also, due to our carelessness earlier, we understated two grants and their numbers earlier, which we have added here. Again, we are so sorry for our carelessness.

4.Please amend the manuscript submission data (via Edit Submission) to include author Yan Xiang and Liangdeng Wang.

Response: We feel sorry for our carelessness. We have added the author Yan Xiang and Liangdeng Wang’s submission data via Edit Submission.

5.Please include captions for your Supporting Information files at the end of your manuscript, and update any in-text citations to match accordingly. 

Response: Supporting information (Page 33, line 761-764) are added at the end of the manuscript. The Supporting information file added is S1 Table. As requested, we updated the citation of the manuscript. Page 29-33, line 562-760.

6.We notice that your supplementary table is uploaded with the file type 'other'. Please amend the file type to 'Supporting Information'. Please ensure that each Supporting Information file has a legend listed in the manuscript after the references list.

Response: We were really sorry for our careless mistakes. We have changed the file type that supplements Table 1 to "Supporting Information". And the document supplementing Table 1 adds a legend after the manuscript's references. Page 33, line 761-764.

7.Please review your reference list to ensure that it is complete and correct. If you have cited papers that have been retracted, please include the rationale for doing so in the manuscript text, or remove these references and replace them with relevant current references. Any changes to the reference list should be mentioned in the rebuttal letter that accompanies your revised manuscript. If you need to cite a retracted article, indicate the article’s retracted status in the References list and also include a citation and full reference for the retraction notice.

Response: According to your suggestion, we have checked all the references in the manuscript and found no such cases. And according to the requirements of your journal, the format of the manuscript and the reference part has been modified.

Responds to the Reviews’ comments: 

1.The authors should include in Table S1 the closest Arabidopsis thaliana homolog for each of the B. rapa genes. This addition would help the average reader identify putative homologs and potential functions of the B. rapa genes more easily.

Response: We sincerely thank the reviewer for careful reading. As suggested by the reviewer, we add the gene name and gene ID of the closest Arabidopsis thaliana homolog for each of the BrJMJ genes to the supplementary file (S1 Table). And upload it to the system as support information.

---

## [Editor Report · Decision Letter 1]

15 Oct 2024

JmjC domain-containing histone demethylase gene family in Chinese cabbage: genome-wide identification and expressional profiling

PONE-D-24-35698R1

Dear Dr. Xu-Feng

We’re pleased to inform you that your manuscript has been judged scientifically suitable for publication and will be formally accepted for publication once it meets all outstanding technical requirements.

Kind regards,

Steven G. Gray

Academic Editor

PLOS ONE

---

## [Editor Report · Acceptance letter]

6 Nov 2024

PONE-D-24-35698R1 

PLOS ONE

Dear Dr. Xiao, 

I'm pleased to inform you that your manuscript has been deemed suitable for publication in PLOS ONE. Congratulations! Your manuscript is now being handed over to our production team.

Kind regards, 

on behalf of

Dr. Steven G. Gray 

Academic Editor

PLOS ONE